# Development and validation of a bedside-available machine learning model to predict discrepancies between SaO$_2$ and SpO$_2$: Exploring factors related to the discrepancies

Raito Sato[1], Naoki Ito[2], Sakina Kadomatsu[3], Norikazu Hanioka[1], Mikio Nakajima[4,5], Tadahiro Goto[6,7]*

1 School of Medical Sciences, University of Fukui, Yoshida, Fukui, Japan, 2 Faculty of Medicine, The University of Tokyo, Bunkyo, Tokyo, Japan, 3 Faculty of Medicine, International University of Health and Welfare, Narita, Chiba, Japan, 4 Department of Clinical Epidemiology and Health Economics, School of Public Health, The University of Tokyo, Bunkyo, Tokyo, Japan, 5 Emergency and Critical Care Center, Tokyo Metropolitan Hiroo Hospital, Shibuya, Tokyo, Japan, 6 Department of Health Data Science, Graduate School of Data Science, Yokohama City University, Yokohama, Japan, 7 TXP Medical Co. Ltd., Chiyoda, Tokyo, Japan

* tag695@mail.harvard.edu

## Abstract

In critically ill patients, a discrepancy frequently exists between percutaneous oxygen saturation (SpO$_2$) and arterial blood oxygen saturation (SaO$_2$), which can lead to potential hypoxemia being overlooked. The aim of this study was to explore the factors related to the discrepancy and to develop an easy-to-use prediction model that uses readily available bedside information to predict the discrepancy and suggest the need for arterial blood gas measurement. This is a prognostic study that used eICU Collaborative Research Database from 2014 to 2015 for model development and MIMIC-IV data from 2008 to 2019 for model validation. To predict the outcome of SpO$_2$ exceeding SaO$_2$ by 3% or more, non-invasive, readily available bedside information (patient demographics, vital signs, vasopressor use, ventilator use) was used to develop prediction models with three machine learning methods (decision tree, logistic regression, XGBoost). To make the model accessible, the model was deployed as a web-based application. Additionally, the contribution of each variable was explored using partial dependence plots and SHAP values. From 4,781 admission records in eICU data, a total of 19,804 paired SpO$_2$ and SaO$_2$ measurements were used. Among three machine learning models, the XGBoost model demonstrated the best predictive performance with an AUROC of 0.73 and a calibration slope of 0.90. In the validation cohort of MIMIC-IV paired dataset, the performance was AUROC of 0.56. An exploratory model-updating step followed by temporal validation raised performance to AUROC of 0.70 with a calibration slope of 0.85. In both datasets, worse vital signs were associated with the discrepancy (e.g., low blood pressure, low temperature) between SpO$_2$ and SaO$_2$. Using non-invasive bedside

**Data availability statement:** All files are available from the MIMIC-IV (https://physionet.org/content/mimiciv/2.2/) and eICU Collaborative Research Database (https://www.nature.com/articles/sdata2018178).

**Funding:** The author(s) received no specific funding for this work.

**Competing interests:** The authors have declared that no competing interests exist.

data, a machine learning model was developed to predict $SpO_2$–$SaO_2$ discrepancy and identified vital signs as key contributors. These findings underscore the awareness for hidden hypoxemia and provide the basis of further study to accurately evaluate the actual $SaO_2$.

## Introduction

The pulse oximeter is essential in clinical settings for non-invasively monitoring percutaneous oxygen saturation ($SpO_2$), instead of a reliable estimate of arterial oxygen saturation ($SaO_2$) typically measured by arterial blood gas (ABG) analysis [1,2]. However, it is well-known that pulse oximeters sometimes overestimate arterial oxygen saturation [2,3]. The accuracy of $SpO_2$ varies due to patient demographics (e.g., race/ethnicity) and conditions (e.g., COVID-19, high HbA1c) [4–8]. The overestimation of $SpO_2$ by pulse oximeters can potentially delay the detection of hypoxemia, especially critical in severe cases, where it may contribute to increased tissue dysfunction and in-hospital mortality [9,10]. Thus, accurately predicting the discrepancy between $SpO_2$ and $SaO_2$ is crucial in the real clinical setting.

Several studies have reported machine learning models that predict partial pressure of oxygen ($PaO_2$) from $SpO_2$ in the patients in the intensive care unit (ICU) [11], but it is limited to them on ventilators. Given that potential hypoxemia can occur in patients not on artificial respiration, it is critically important to identify inaccuracies of pulse oximeters across various clinical settings. In addition, factors associated with the discrepancies were not well-determined.

This study aimed to develop and validate a model to predict the discrepancies between $SpO_2$ and $SaO_2$ using simple, non-invasive bedside information. This model is designed to predict pulse oximeter overestimations, inform the need for ABG analysis, and aid in preventing missed diagnoses of potential hypoxemia. The contribution of each variable to the discrepancy was further investigated.

## Materials and methods

### Study design and settings

This is a prognostic study that uses two publicly available patient-level ICU databases: the eICU Collaborative Research Database (eICU) version 2.0 and the Medical Information Mart for Intensive Care IV (MIMIC-IV) dataset version 2.2. The eICU comprises data from over 13,000 ICU patients were admitted to one of 335 units at 208 hospitals located throughout the United States, spanning from 2014 to 2015 [12]. MIMIC-IV, an extensive database from the Beth Israel Deaconess Medical Center in the United States, includes de-identified health-related data of over 60,000 patients, admitted from 2008 to 2019 [13]. The eICU database was used for model development, and the MIMIC-IV database for both model fine-tuning and validation. The eICU database was accessed on March 20, 2024, and the MIMIC-IV database was accessed on September 1, 2023. Ethical approval and informed consent were deemed unnecessary for this study as eICU and MIMIC-IV data are de-identified in

accordance with HIPAA's Safe Harbor provision, and data access was restricted to credentialed authors who complied with the specified data use agreement. Consequently, the TXP Medical Ethical Review Board waived the requirement for ethical approval and informed consent (TXPREC-008). This study adhered to the Transparent Reporting of a multivariable prediction model for Individual Prognosis or Diagnosis (TRIPOD) guidelines for prognostic studies [14].

## Study samples

This study included adult patients (aged ≥18 years), targeting individuals identified as Black, White, Hispanic, or Asian. Measurements required $SpO_2$ and $SaO_2$ levels to be recorded within 10 minutes of each other, with $SpO_2$ ranging from 80% to 100% and $SaO_2$ from 50% to 100%. The difference between $SpO_2$ and $SaO_2$ in this study was required to be no more than 20%. The following criteria were established for inclusion: mean blood pressure (MBP) should range from 50 to 180 mm Hg, respiratory rate (RR) should range from 5 to 50 breaths per minute, heart rate (HR) should range from 30 to 140 beats per minute, body mass index (BMI) should range from 15 to 50 kg/m², and body temperature should range from 32 to 42°C. Additionally, MBP, RR, and HR measurements were needed within 10 minutes of the $SaO_2$ reading, and body temperature should be recorded within 3 hours of the $SaO_2$ measurement. Any records with missing data were excluded from this analysis.

## Predictors

The predictors for machine learning models were chosen from routinely available data at ICU. Specifically, the predictors included patient age, sex, BMI, race, vital signs (temperature, HR, MBP, RR, and $SpO_2$), the use of invasive ventilation (yes/no), and the use of vasopressor (yes/no).

## Outcomes

The outcome of the study was $SpO_2$ being 3% higher than $SaO_2$

## Statistical analysis

In the training set (80% random sample), the risk of algorithmic bias was reduced by applying and comparing three machine learning methods: (1) Decision Tree, (2) Logistic Regression, and (3) eXtreme Gradient Boosting (XGBoost). Model performance was assessed in the training set, with the area under the receiver operating characteristic curve (AUROC) as the optimization metric, and 95% confidence intervals (CI) for AUROC were calculated. Hyperparameter tuning for each model was performed using a grid search combined with 10-fold cross-validation. The specific parameter ranges and selected values for XGBoost are provided in S2 Table. Final model evaluation was conducted on the remaining 20% test set, using multiple diagnostic metrics including AUROC, sensitivity, specificity, positive likelihood ratio (Positive LR), negative likelihood ratio (Negative LR), and Diagnostic Odds Ratio. The model with the highest overall diagnostic value was selected for further fine-tuning and validation, and showed its calibration plot and calibration slope. The calibration plot visually illustrates the agreement between predicted probabilities and actual outcomes, while the calibration slope quantifies the degree of calibration. These visual and numerical assessments provide insights into the model's calibration performance. As for exploratory analysis to investigate whether performance could be improved in the new independent dataset, this study conducted an exploratory hyper-parameter optimization using admissions from 2008–2013 and then evaluated the updated model on the held-out 2014–2019 data. In this study, two distinct techniques for model interpretation were employed: SHapley Additive exPlanations (SHAP) analysis, and Partial Dependency Plots (PDPs). Both methods demonstrate the impact of different features on the model's predictive output. These analyses, visualizations, and the model publication were conducted using R (version 4.2.2), Python (version 3.10.12), and the Python library Streamlit (version 1.26.0).

## Results

During 2014–2015, the eICU database recorded 70,304 data points containing both $SpO_2$ and $SaO_2$ measurements. Data points were excluded for the following reasons: 44,330 had significant time gaps over 10 minutes between $SpO_2$ and $SaO_2$ measurements, 4,789 showed outliers, and 1,381 were missing data. Finally, 4,781 admissions and 19,804 data points were identified as suitable for analysis. There were 5,231 data points where $SpO_2$ exceeded $SaO_2$ by 3% (Tables 1 and 2). In the validation cohort using the MIMIC-IV database (2008–2019), there were 4,267 admissions and 9,339 data points. Of these, 5,953 data points were from 2008 to 2013, and 3,386 data points were from 2014 to 2019. Study flows are shown in Fig 1.

### Comparison of three training models

Among the three models compared, the XGBoost model demonstrated the highest predictive performance on the test set (20% of dataset) with AUC values of 0.73 (95% CI: 0.71–0.74) for XGBoost, 0.60 (95% CI: 0.58–0.62) for Decision Tree model, and 0.57 (95% CI: 0.55–0.59) for Logistic Regression (S1 Fig). After comparing other evaluation scores (S1 Table), the XGBoost model was selected for prediction. Additionally, a calibration plot was constructed to further assess the model's performance. The calibration slope was 0.90 (S2 Fig).

### SHAP and partial dependence plot

The SHAP summary plot revealed the top 14 features identified by the XGBoost model, ranked by their average SHAP values. These values indicate the positive or negative impact of each feature (Fig 2). To further elucidate the associations, PDPs were created, focusing on the relationship between six key variables (BMI, Age, HR, Temperature, MBP, RR) and the discrepancy in pulse oximetry measurements (Fig 2). Among these variables, Age, HR and RR showed increasing trends in association with the discrepancy, whereas Temperature and MBP displayed inverse trends.

**Table 1. Predictor variables and outcome in 19,804 data from eICU.**

| Variable | Mean±SD | Median (IQR) | Number (%) |
|---|---|---|---|
| Age, years | 61.8±15.5 | 64.0 (53.0–73.0) | |
| Female | | | 7,128 (36.0%) |
| BMI | 29.7±6.6 | 29.0 (25.0–33.8) | |
| Race | | | |
| White | | | 17,063 (86.2%) |
| Black | | | 1,660 (8.4%) |
| Hispanic | | | 560 (2.8%) |
| Asian | | | 521 (2.6%) |
| Vital signs | | | |
| Temperature,°C | 36.9±1.2 | 37.0 (36.4–37.6) | |
| Heart rate, beat/min | 87.2±17.7 | 86.0 (75.0–98.0) | |
| Mean blood pressure, mm Hg | 78.6±15.6 | 76.0 (68.0–86.0) | |
| Respiratory rate, breaths/min | 18.7±6.3 | 18.0 (14.0–22.0) | |
| Oxygen saturation, % | 97.5±3.2 | 99.0 (96.0–100.0) | |
| Treatment | | | |
| Invasive ventilation | | | 14,146 (71.4%) |
| Vasopressor | | | 7,660 (38.7%) |
| Outcome | | | |
| $SpO_2$–$SaO_2$ discrepancy ≥3% | | | 5,231 (26.4%) |

eICU = eICU Collaborative Research Database, IQR = Interquartile range, SD = Standard deviation

**Table 2. Patient characteristics stratified by SpO$_2$–SaO$_2$ discrepancy ≥3%.**

| Variable | Discrepancy ≥3% (n = 5,231) | Discrepancy <3% (n = 14,573) |
|---|---|---|
| Age, median (IQR), years | 64.0 (54.0-72.0) | 64.0 (53.0-73.0) |
| Female, number (%) | 1,947 (37.2%) | 5,181 (35.6%) |
| BMI, median (IQR) | 29.0 (25.0-33.8) | 29.0 (24.9-33.8) |
| Race, number (%) | | |
| White | 4,337 (82.9%) | 12,726 (87.3%) |
| Black | 508 (9.7%) | 1,152 (7.9%) |
| Hispanic | 165 (3.2%) | 395 (2.7%) |
| Asian | 221 (4.2%) | 300 (2.1%) |
| Vital signs, median (IQR) | | |
| Temperature,°C | 37.0 (36.3-37.5) | 37.0 (36.4-37.6) |
| Heart rate, beat/min | 87.0 (75.0-99.0) | 86.0 (75.0-98.0) |
| Mean blood pressure, mm Hg | 75.0 (68.0-85.0) | 76.0 (69.0-86.0) |
| Respiratory rate, breaths/min | 18.0 (14.0-23.0) | 18.0 (14.0-22.0) |
| Oxygen saturation, % | 99.0 (96.0-100.0) | 98.0 (96.0-100.0) |
| Treatment, number (%) | | |
| Invasive ventilation | 3,765 (72.0%) | 10,381 (71.2%) |
| Vasopressor | 1,963 (37.5%) | 5,697 (39.1%) |

IQR = Interquartile range.

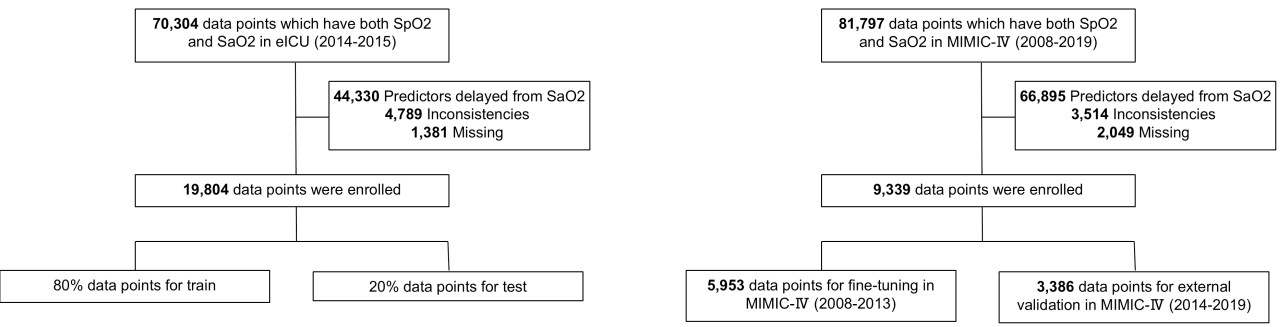

**(A) The development cohort**

**(B) The fine-tuning and validation cohort**

**Fig 1. Flow diagram of patients eligible for analysis in the development and validation cohorts.** (A) In the eICU dataset (2014–2015), 70,304 data points included both SpO$_2$ and SaO$_2$ measurements. Among these, 19,804 data points also contained all other predictors (i.e., patient age, sex, BMI, race, temperature, HR, MBP, RR, the use of invasive ventilation, and the use of vasopressor). (B) In the MIMIC-IV dataset (2008–2019), 81,797 data points included both SpO$_2$ and SaO$_2$ measurements. Among these, 9,350 data points also contained all other predictors. SpO$_2$ = Percutaneous Oxygen Saturation, SaO$_2$ = Arterial Oxygen Saturation, eICU = eICU Collaborative Research Database, MIMIC-IV = Medical Information Mart for Intensive Care IV, BMI = Body mass index, HR = Heart rate, MBP = Mean blood pressure, RR = Respiratory rate.

## Fine-tuning and temporal validation

The original XGBoost model was first validated on the MIMIC-IV dataset, which yielded an AUROC of 0.56. After fine-tuning the XGBoost model on the earlier years of MIMIC-IV (2008–2013), and validating it on the remaining years of MIMIC-IV (2014–2019), the AUROC was 0.70 (95% CI: 0.68–0.72), and the calibration slope is 0.85 (S3 Fig). SHAP values in the validation cohort showed a similar pattern to those observed in the training cohort (Fig 3). Additionally, PDPs, shown in the same figure, exhibited similar trends to those in the internal validation, indicating that vital signs have an influence on prediction.

**(A) SHAP Value in the development cohort**

**(B) PDPs in the development cohort**

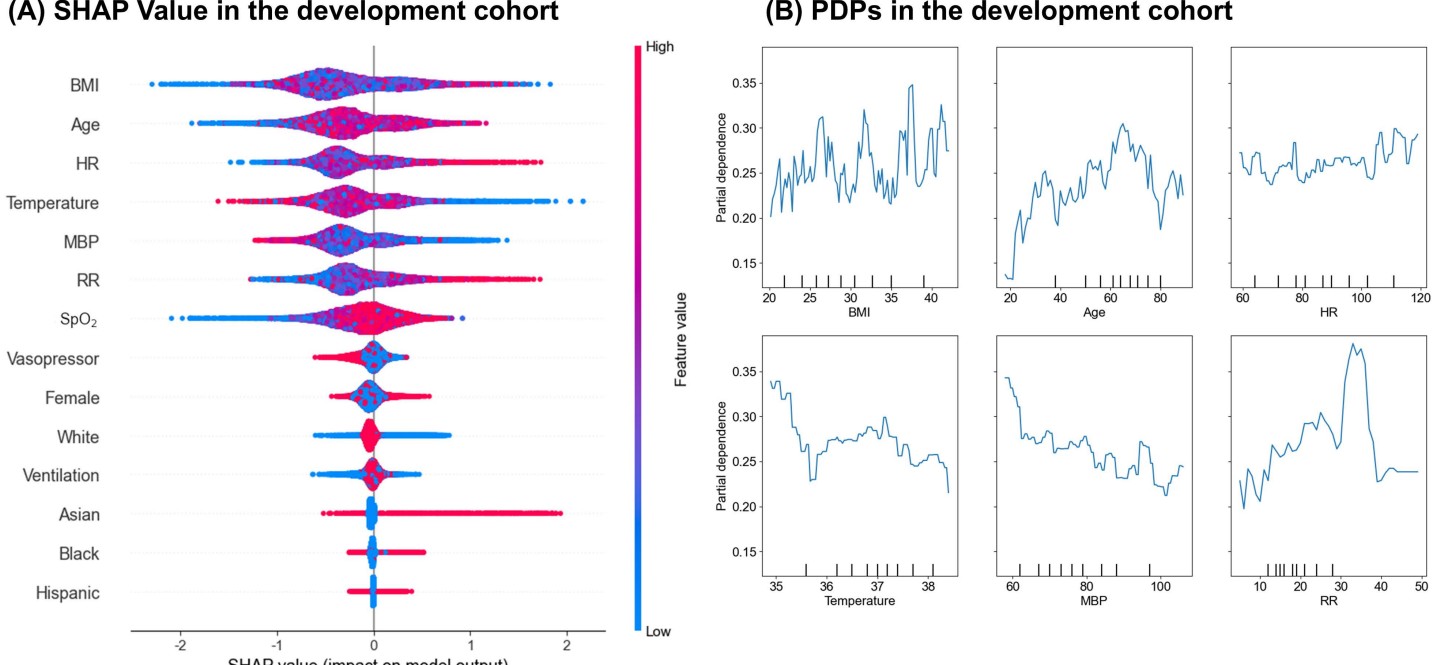

**Fig 2. SHAP value and PDPs in the development cohort.** (A) SHAP value plot showing the impact of each predictor on the model's output in the development cohort using the eICU dataset. Each point represents a single data point, with the color indicating the feature's value (red for high values and blue for low values). The position on the x-axis shows the effect of the feature on the prediction, with values to the right indicating a higher predicted outcome. (B) Partial Dependence Plots (PDPs) for key predictors in the development cohort using the eICU dataset. Each plot displays the relationship between a specific predictor and the predicted outcome, while averaging out the effects of all other predictors in the model. The x-axis represents the range of the predictor's values, and the y-axis shows the average predicted outcome, providing insight into how changes in the predictor's value influence the model's predictions. SHAP = SHapley Additive exPlanations, eICU = eICU Collaborative Research Database, MIMIC-IV = Medical Information Mart for Intensive Care IV.

## Discussion

In this study, a machine learning model was developed using 19,804 data points from the eICU database to accurately predict overestimations made by pulse oximeters. In both datasets, worse vital signs were associated with the $SpO_2$–$SaO_2$ discrepancy. For example, low body temperature, low MBP, and high RR were associated with the $SpO_2$–$SaO_2$ discrepancy, indicating that, in the ICU, there is a need to pay attention to the discrepancy, particularly in critically ill patients.

Compared with a previously reported model that achieved an AUROC of about 0.83 for estimating $PaO_2$ from $SpO_2$ in ventilated patients [10], this model shows lower discrimination. Its exclusive reliance on non-invasive inputs may partly explain the reduced precision in estimating $SaO_2$. Even so, an exploratory temporal-validation analysis produced an AUROC of 0.70, suggesting reasonably stable performance in unseen patients and offering a clinically relevant gauge of generalizability. Further gains may be possible by expanding the training dataset and fine-tuning the timing of the paired measurements.

Interestingly, the SHAP analysis showed that the top-ranked predictors were consistent in the training (eICU) and validation (MIMIC) datasets, indicating that these variables retain their importance across settings and may support wider generalizability. Partial-dependence plots for heart rate, mean arterial pressure, and respiratory rate displayed similar risk gradients in both cohorts, illustrating how changes in vital signs affect the predicted risk and reducing the model's black-box perception for clinicians. The agreement between SHAP values and PDP trends suggests that these routinely recorded clinical features are key drivers of the predictions.

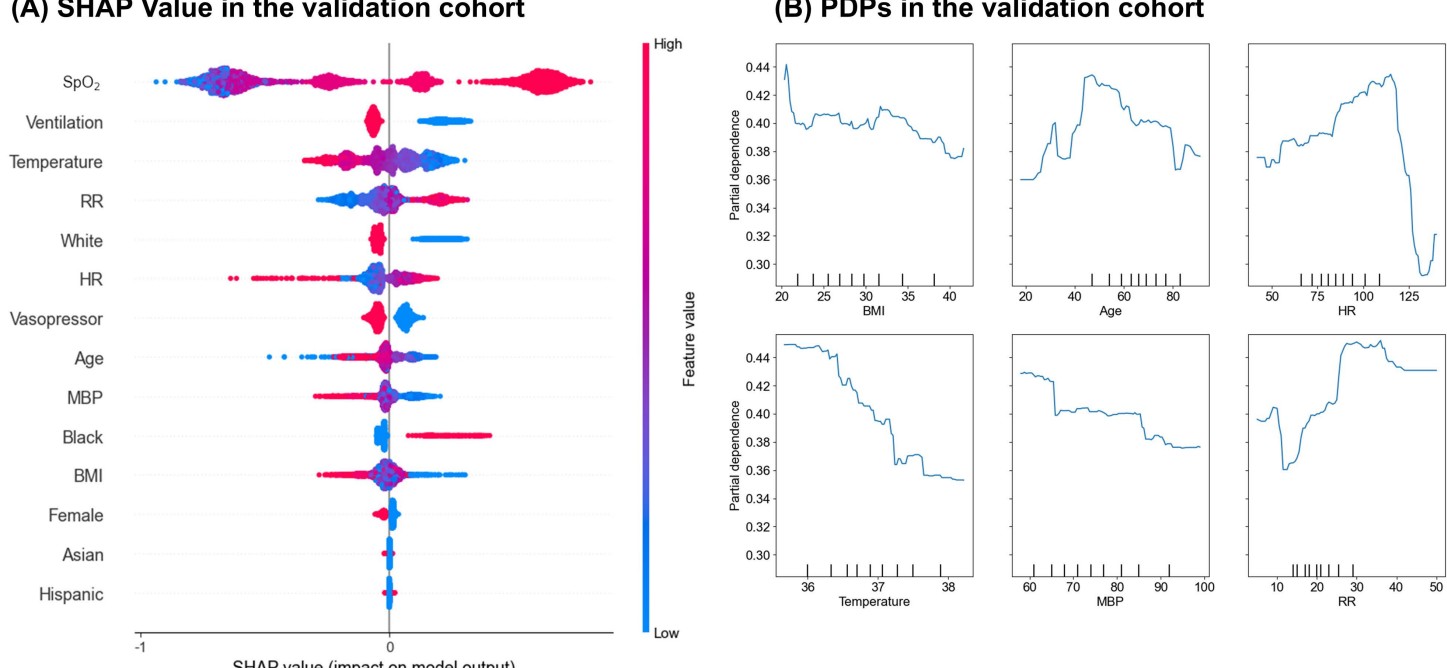

**Fig 3. SHAP value and PDPs in the validation cohort.** (A) SHAP value plot for the validation cohort using the MIMIC-IV dataset. The model developed with the eICU data (development cohort) was applied to the MIMIC-IV (2014-2019) to assess its generalizability. Each point represents a single data point, with the color indicating the feature's value. The position on the x-axis shows the effect of the feature on the prediction, with values to the right indicating a higher predicted outcome. (B) Partial Dependence Plots (PDPs) for key predictors in the validation cohort using the MIMIC-IV (2014-2019) dataset. The PDPs are aligned with those used in the development cohort (eICU) to ensure consistency in comparison. Each plot displays the relationship between a specific predictor and the predicted outcome, while averaging out the effects of all other predictors in the model. The x-axis represents the range of the predictor's values, and the y-axis shows the average predicted outcome, providing insight into how changes in the predictor's value influence the model's predictions. SHAP = SHapley Additive exPlanations, eICU = eICU Collaborative Research Database, MIMIC-IV = Medical Information Mart for Intensive Care IV.

These findings about the correlation between the worsening vital signs (e.g., low temperature and low blood pressure) leading to low perfusion and the decrease in pulse oximeter accuracy are consistent with previous studies. For example, when patients have low perfusion, the accuracy of the pulse oximeter decreases [15,16]. Additionally, the accuracy of the pulse oximeter is influenced by the patient's body temperature, often leading to overestimated readings when the body temperature is low [17].

A key strength of this study lies in offering clear insights into how various factors affect discrepancies between SpO$_2$ and SaO$_2$, simplifying the understanding of these complex interactions. These factors may influence on overlooking patients with hypoxemia, which is associated with poor prognosis. This study emphasizes the necessity of vigilant monitoring for hidden hypoxemia in critically ill patients. It also contributes to the ongoing development of accurate, non-invasive methods for assessing oxygenation status, marking a significant step forward in improving patient care in critical care environments.

This model demonstrated stable predictive performance in a temporally split validation using the MIMIC-IV dataset, suggesting its potential for integration into clinical workflows as a decision-support tool. In particular, it may help identify patients at risk of hypoxemia, especially those with low perfusion or other critical vital signs. The model outputs can be used to guide the prioritization of patient monitoring in ICU settings, while still relying on clinical judgment and confirmatory tests such as blood gas analysis. Clinicians can use the model's predictions to support early recognition and intervention for patients who may require closer observation.

## Model implementation

A lightweight, publicly accessible web application (https://spo2-to-sao2.streamlit.app/) implementing this prediction model has been released. Users manually enter readily available information—vital signs and basic patient characteristics—and the app instantly returns the estimated probability of an $SpO_2$–$SaO_2$ discrepancy (≥ 3 percentage points). Although the tool is not yet embedded in hospital information systems and therefore requires hand entry, it still allows clinicians to gauge risk at the bedside and may serve as an engaging proof-of-concept for future, fully integrated deployments.

## Potential limitation

This study has several limitations. First, $SaO_2$-predictor pairs combine measurements that are not always collected simultaneously. For a more accurate analysis, the data were restricted to vital signs and ABG tests that were recorded within 10 minutes. Second, the database used for training in this study is solely from the United States, which may limit the generalizability of these results. However, to address this concern and enhance the robustness of these findings, external validation was conducted. Moreover, another limitation of this study is the potential bias arising from using multiple data points from the same patients. This aspect is critical in interpreting the data and suggests the need for future research to consider individual patient variability more thoroughly.

## Conclusion

By using non-invasive, readily available bedside information, a machine learning model was developed to predict when $SpO_2$ exceeds $SaO_2$ by 3% or more, while the prediction ability was suboptimal in a different dataset. Vital signs (e.g., temperature and heart rate) were identified as factors associated with these discrepancies. These findings underscore the need for awareness of hidden hypoxemia and provide a basis of further studies to identify hidden-hypoxia in critically ill patients.

## Supporting information

**S1 Fig. Comparison of three machine models ROC curves and AUC values.** This figure shows the ROC curves and AUC values for three machine learning models: (1) Decision Tree, (2) Logistic Regression, and (3) XGBoost. The ROC curves illustrate each model's performance by plotting the true positive rate against the false positive rate. The AUC values, displayed on each curve, indicate the overall performance of the models, with higher values representing better discriminatory ability. ROC = Receiver operating characteristic, AUC = Area under the curve, XGBoost = eXtreme Gradient Boosting.
(TIF)

**S1 Table. Three models' scores.** AUROC = Area under the receiver operating characteristic curve, XGBoost = eXtreme Gradient Boosting, Positive LR = Positive Likelihood Ratio, Negative LR = Negative Likelihood Ratio, Diagnostic OR = Diagnostic Odds Ratio.
(TIF)

**S2 Table. Hyperparameter search space for the XGBoost model.** XGBoost = eXtreme Gradient Boosting.
(TIF)

**S2 Fig. Calibration plot and calibration slope in the development cohort.** This figure presents the calibration plot and calibration slope for the development cohort using the eICU data set. The calibration plot compares the predicted probabilities of the model with the actual outcomes, illustrating how well the model's predictions match the observed results. The ideal line represents perfect calibration, where predicted probabilities exactly match the observed frequencies. The calibration slope indicates the agreement between predicted probabilities and actual outcomes. A slope of 1 suggests

perfect calibration, while deviations from 1 indicate under- or over-estimation of the predicted risks. eICU = eICU Collaborative Research Database.
(TIF)

**S3 Fig. ROC curve and calibration plot in the validation cohort.** This figure presents the ROC curve and calibration plot for the validation cohort using the MIMIC-IV dataset (2014–2019). The model was trained using data from the eICU database and validated using the MIMIC-IV dataset (2014–2019). ROC = Receiver Operating Characteristic, MIMIC-IV = Medical Information Mart for Intensive Care IV, eICU = eICU Collaborative Research Database.
(TIF)

## Author contributions

**Conceptualization:** Tadahiro Goto.

**Data curation:** Raito Sato, Naoki Ito.

**Formal analysis:** Raito Sato, Naoki Ito, Norikazu Hanioka.

**Investigation:** Raito Sato, Naoki Ito, Sakina Kadomatsu, Norikazu Hanioka.

**Project administration:** Tadahiro Goto.

**Software:** Raito Sato.

**Supervision:** Mikio Nakajima, Tadahiro Goto.

**Validation:** Raito Sato.

**Visualization:** Raito Sato, Sakina Kadomatsu.

**Writing – original draft:** Raito Sato.

**Writing – review & editing:** Mikio Nakajima, Tadahiro Goto.

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
