## [Decision Letter · Decision Letter 0]

11 Mar 2025

PONE-D-24-58502Development and Validation of a Bedside-Available Machine Learning Model to Predict Discrepancies between SaO₂ and SpO₂: Exploring Factors Related to the DiscrepanciesPLOS ONE

Dear Dr. Goto,

Thank you for submitting your manuscript to PLOS ONE. After careful consideration, we feel that it has merit but does not fully meet PLOS ONE’s publication criteria as it currently stands. Therefore, we invite you to submit a revised version of the manuscript that addresses the points raised during the review process.

We look forward to receiving your revised manuscript.

Kind regards,

Mohsen Mehrabi

Academic Editor

PLOS ONE

Journal Requirements:

1. Please ensure that your manuscript meets PLOS ONE's style requirements, including those for file naming. The PLOS ONE style templates can be found at https://journals.plos.org/plosone/s/file?id=wjVg/PLOSOne_formatting_sample_main_body.pdf and https://journals.plos.org/plosone/s/file?id=ba62/PLOSOne_formatting_sample_title_authors_affiliations.pdf.   

3. "We notice that your supplementary figures are uploaded with the file type 'Figure'. Please amend the file type to 'Supporting Information'. Please ensure that each Supporting Information file has a legend listed in the manuscript after the references list.

4. Thank you for uploading your study's underlying data set. Unfortunately, the repository you have noted in your Data Availability statement does not qualify as an acceptable data repository according to PLOS's standards.

Additional Editor Comments:

Thank you for your submission titled "Development and Validation of a Bedside-Available Machine Learning Model to Predict Discrepancies between SaO₂ and SpO₂: Exploring Factors Related to the Discrepancies." We appreciate your efforts in addressing an important issue in critical care.

After careful consideration and review, we have received feedback from two reviewers. The key points are as follows:

1- Reviewer 1 has suggested Minor Revisions, indicating that while your manuscript is generally well-prepared, there are a few areas that require clarification and improvement. We encourage you to carefully address their comments to enhance the overall quality of your work.

2- Reviewer 2 has recommended Major Revisions, citing significant concerns that must be addressed before the manuscript can be considered for publication. It is crucial that you thoroughly respond to each of the points raised by this reviewer to ensure the validity and robustness of your findings.

In your revision, please provide a detailed response to each comment from the reviewers, indicating how you have addressed their suggestions or explaining any points of disagreement.

We look forward to receiving your revised manuscript by April 09, 2025. Thank you for your contribution to our journal.

Reviewers' comments:

Reviewer's Responses to Questions

**Comments to the Author**

1. Is the manuscript technically sound, and do the data support the conclusions?

Reviewer #1: Yes

Reviewer #2: Yes

2. Has the statistical analysis been performed appropriately and rigorously? 

Reviewer #1: Yes

Reviewer #2: Yes

3. Have the authors made all data underlying the findings in their manuscript fully available?

Reviewer #1: Yes

Reviewer #2: Yes

4. Is the manuscript presented in an intelligible fashion and written in standard English?

Reviewer #1: Yes

Reviewer #2: Yes

5. Review Comments to the Author

Reviewer #1: This study presents a methodologically rigorous and well-structured investigation employing two distinct publicly available datasets. The primary dataset was utilized to develop a machine learning model aimed at predicting instances where pulse oximetry might overestimate arterial oxygen saturation. Subsequently, the optimal model was applied to a secondary dataset, yielding significantly lower performance metrics. This finding underscores the challenges associated with model generalization across heterogeneous datasets.

The publication of studies reporting negative results is of paramount importance for scientific advancement. This work makes a valuable contribution by highlighting the necessity of prudence when extrapolating machine learning models to external datasets not included in the training phase.

While the methodological framework is robust, additional details would enhance the transparency and reproducibility of the study. In particular, I recommend the following clarifications:

(1) The manuscript states that the training phase was conducted using 80% of the dataset. However, it remains unclear how the remaining 20% sample was utilized. A precise description of its role in validation or testing would be beneficial.

(2) The reported AUC values require further clarification. Specifically, it should be explicitly stated whether they represent the mean AUC obtained during cross-validation or the AUC computed on the 20% test sample.

(3) A detailed explanation of the hyperparameter tuning process, particularly for the XGBoost model, is warranted. The absence of Table S1 in the provided version of the manuscript prevents verification of whether these details are available. If already included in the supplementary materials, this point may be disregarded.

Overall, this manuscript constitutes a valuable addition to the field. Addressing the aforementioned points would strengthen its clarity and methodological rigor. I commend the authors for their efforts and look forward to seeing the revised version of this work.

Reviewer #2: Strengths:

The topic is highly relevant for critical care settings, addressing the discrepancy between SpO₂ and SaO₂, which has significant clinical implications.

The study utilizes real-world datasets (eICU and MIMIC-IV) for both model development and validation.

The exploration of SHAP values and partial dependence plots helps interpret model predictions.

Major Issues:

Generalizability and Model Performance: The model performs well on the training data (AUROC: 0.73) but poorly on the validation set (AUROC: 0.56). This indicates potential overfitting or dataset bias, which must be addressed.

Feature Importance Analysis: While SHAP values are mentioned, there is limited discussion on how these insights could be used to improve the model.

Clinical Utility: The paper should clarify how the model can be integrated into clinical workflows. Would it be used as a decision-support tool? How should clinicians interpret the results?

Web-Based Implementation Details: The web-based tool mentioned is not sufficiently described. A brief description or a link to a demonstration would be beneficial.

Statistical Analysis: More robust statistical validation, such as confidence intervals for AUROC, would strengthen the claims.

6. PLOS authors have the option to publish the peer review history of their article (what does this mean? ). If published, this will include your full peer review and any attached files.

**Do you want your identity to be public for this peer review?** For information about this choice, including consent withdrawal, please see our Privacy Policy .

Reviewer #1: **Yes: ** Christian Valle Morinaga

Reviewer #2: No

---

## [Author Response · Author response to Decision Letter 1]

24 Apr 2025

Editorial Office

PLOS ONE

Dear Dr. Mohsen Mehrabi and Reviewers,

We sincerely appreciate the time and effort that you and the reviewers have dedicated to evaluating our manuscript, "Development and Validation of a Bedside-Available Machine Learning Model to Predict Discrepancies between SaO₂ and SpO₂: Exploring Factors Related to the Discrepancies." We are grateful for the constructive comments and suggestions, which have helped us improve our manuscript. Below, we provide a detailed point-by-point response to the reviewers' comments, along with the corresponding revisions made to the manuscript.

Reviewer 1

Comment 1:

The manuscript states that the training phase was conducted using 80% of the dataset. However, it remains unclear how the remaining 20% sample was utilized. A precise description of its role in validation or testing would be beneficial.

Response:

Thank you for your comment. The dataset was randomly split into 80% for training and 20% for testing. During the training phase, 10-fold cross-validation was conducted exclusively within the 80% training set to optimize hyperparameters and assess model performance. The final model was then evaluated using the remaining 20% as an independent test set. We have clarified this in the manuscript.

Revision:

The Statistical analysis section (page 8, para 1) has been updated to specify that 80% of the dataset was used for training with cross-validation, while the remaining 20% was used as an independent test set.

“In the training set (80% random sample), we reduced the risk of algorithmic bias by applying and comparing three machine learning methods, … , Final model evaluation was conducted on the remaining 20% test set, using multiple diagnostic metrics including AUROC, sensitivity, specificity, positive likelihood ratio (Positive LR), negative likelihood ratio (Negative LR), and Diagnostic Odds Ratio”

Comment 2:

The reported AUC values require further clarification. Specifically, it should be explicitly stated whether they represent the mean AUC obtained during cross-validation or the AUC computed on the 20% test sample.

Response:

During model training, 10-fold cross-validation was performed within the 80% training set, and the mean AUC from cross-validation was used for model selection. The final model was then evaluated on the independent 20% test set, and the AUC on this test set was separately reported. We have clarified this distinction in the manuscript.

Revision:

Clarifications regarding test set AUC have been added to the Results section, Comparison of three training models subsection (page 11, para 2) as follows: “Among the three models compared, the XGBoost model demonstrated the highest predictive performance on the test set (20% of dataset) with AUC values of 0.73”

Comment 3:

A detailed explanation of the hyperparameter tuning process, particularly for the XGBoost model, is warranted. The absence of Table S1 in the provided version of the manuscript prevents verification of whether these details are available.

Response:

Thank you for your suggestion. To enhance clarity, we have added a new supplementary table (Table S2) that provides a detailed overview of the hyperparameter tuning process for XGBoost, including the search ranges and final values selected. The manuscript now explicitly refers to this table.

Revision:

The Statistical Analysis section (page 8, para 1) has been updated to include a description of the hyperparameter tuning process for XGBoost. Additionally, a new supplementary table (Table S2) has been added to present the specific parameters tested and their search ranges. This information is now explicitly referenced in the manuscript.

“Hyperparameter tuning for each model was performed using a grid search combined with 10-fold cross-validation. The specific parameter ranges and selected values for XGBoost are provided in Table S2.”

Reviewer 2:

Comment 1:

Generalizability and Model Performance: The model performs well on the training data (AUROC: 0.73) but poorly on the validation set (AUROC: 0.56). This indicates potential overfitting or dataset bias, which must be addressed.

Response:

Thank you for your thorough review. As you pointed out, the pre-specified model demonstrated limited transportability on the full external cohort. To investigate whether performance could be improved, we conducted an exploratory hyper-parameter optimization using admissions from 2008–2013 and then evaluated the updated model on the held-out 2014–2019 data. We present this procedure as model updating plus temporal validation and, for transparency, continue to report the original AUROC. The updated model achieved an AUROC of 0.70 (95 % CI 0.68–0.73) with a calibration slope of 0.85. Because the predictor set is restricted to routinely collected variables, the model can be updated in most hospitals with minimal effort. Nevertheless, these findings remain exploratory; we have added a statement noting that additional validation in independent centers and geographical regions is planned and still required.

Revision:

The Methods section (page 8, para 1) “As for exploratory analysis to investigate whether performance could be improved in the new independent dataset, we further conducted an exploratory hyper-parameter optimization using admissions from 2008–2013 and then evaluated the updated model on the held-out 2014–2019 data.”

The Discussion section (page 15, para 2): “Compared with a previously reported model that achieved an AUROC of about 0.83 for estimating PaO₂ from SpO₂ in ventilated patients [10], our model shows lower discrimination. Its exclusive reliance on non-invasive inputs may partly explain the reduced precision in estimating SaO₂. Even so, an exploratory temporal-validation analysis produced an AUROC of 0.70, suggesting reasonably stable performance in unseen patients and offering a clinically relevant gauge of generalizability. Further gains may be possible by expanding the training dataset and fine-tuning the timing of the paired measurements.”

Comment 2:

Feature Importance Analysis: While SHAP values are mentioned, there is limited discussion on how these insights could be used to improve the model.

Response:

Thank you for your valuable comment. We have expanded the discussion on SHAP values, emphasizing the consistency of top-ranking features across both the training (eICU) and validation (MIMIC) datasets. This consistency strengthens the generalizability of the model. Additionally, we have elaborated on how these insights, supported by similar trends in PDPs, contribute to both model performance and clinical decision-making.

Revision:

In the Discussion section (page 16, para 1), “SHAP analysis showed that the top-ranked predictors were consistent in the training (eICU) and validation (MIMIC) datasets, indicating that these variables retain their importance across settings and may support wider generalizability. Partial-dependence plots for heart rate, mean arterial pressure, and respiratory rate displayed similar risk gradients in both cohorts, illustrating how changes in vital signs affect the predicted risk and reducing the model’s black-box perception for clinicians. The agreement between SHAP values and PDP trends suggests that these routinely recorded clinical features are key drivers of the predictions.”

Comment 3:

Clinical Utility: The paper should clarify how the model can be integrated into clinical workflows. Would it be used as a decision-support tool? How should clinicians interpret the results?

Response:

Thank you for your comment. We have clarified in the discussion that the model can be used as a decision-support tool in ICU settings to help prioritize patients at risk for hypoxemia. The model’s output should guide clinicians in identifying patients who may need closer monitoring.

Revision:

We added to the Discussion section (page 17, para2) “Our model demonstrated stable predictive performance in a temporally split validation using the MIMIC-IV dataset, suggesting its potential for integration into clinical workflows as a decision-support tool. In particular, it may help identify patients at risk of hypoxemia, especially those with low perfusion or other critical vital signs. The model outputs can be used to guide the prioritization of patient monitoring in ICU settings, while still relying on clinical judgment and confirmatory tests such as blood gas analysis. Clinicians can use the model’s predictions to support early recognition and intervention for patients who may require closer observation.

”

Comment 4:

Web-Based Implementation Details: The web-based tool mentioned is not sufficiently described. A brief description or a link to a demonstration would be beneficial.

Response:

Thank you for your valuable feedback. We have now added a brief description of the web-based tool and its functionality, clarifying how clinicians can input key predictors to obtain a probability estimate for SpO₂–SaO₂ discrepancy.

Revision:

The Model implementation section (page 17, para 3) “We have released a lightweight, publicly accessible web application (https://spo2-to-sao2.streamlit.app/) that implements our prediction model. Users manually enter readily available information—vital signs and basic patient characteristics—and the app instantly returns the estimated probability of an SpO₂–SaO₂ discrepancy (≥ 3 percentage points). Although the tool is not yet embedded in hospital information systems and therefore requires hand entry, it still allows clinicians to gauge risk at the bedside and may serve as an engaging proof-of-concept for future, fully integrated deployments.”

Comment 5:

Statistical Analysis: More robust statistical validation, such as confidence intervals for AUROC, would strengthen the claims.

Response:

Thank you for your recommendation. We have now included confidence intervals for AUROC values in both the internal and external validation results.

We believe that these revisions have strengthened our manuscript and addressed the concerns raised by the reviewers. We sincerely appreciate the reviewers’ thoughtful comments and the opportunity to improve our work. Please let us know if any further clarifications are required.

Best regards,

Raito Sato

School of Medical Sciences, University of Fukui

---

## [Decision Letter · Decision Letter 1]

17 Jul 2025

PONE-D-24-58502R1Development and Validation of a Bedside-Available Machine Learning Model to Predict Discrepancies between SaO₂ and SpO₂: Exploring Factors Related to the DiscrepanciesPLOS ONE

Dear Dr. Goto,

Thank you for submitting your manuscript to PLOS ONE. After careful consideration, we feel that it has merit but does not fully meet PLOS ONE’s publication criteria as it currently stands. Therefore, we invite you to submit a revised version of the manuscript that addresses the points raised during the review process.

**ACADEMIC EDITOR: **

Required for acceptance:

– Enhance the descriptive table on page 10 by adding more detailed statistics (e.g., means, SD, ranges) and include a bivariate table if the outcome is binary, per Reviewer 3.

– Improve English as noted by Reviewer 2 (e.g., avoid “we”/“I”).

With these minor revisions, the paper meets all PLOS ONE technical and reporting criteria and can be accepted without further review.

We look forward to receiving your revised manuscript.

Kind regards,

Mohsen Mehrabi, Ph.D.

Academic Editor

PLOS ONE

Journal Requirements:

Additional Editor Comments :

Thank you for your submission. Two reviewers recommended acceptance, and one requested a minor revision concerning the descriptive statistics table on page 10. Please improve the table by adding more detailed statistics as suggested. Also, kindly revise the language as per Reviewer 2's comment regarding the use of pronouns like "we" or "I." Once these minor revisions are addressed, the manuscript will be accepted without further review.

Reviewers' comments:

Reviewer's Responses to Questions

**Comments to the Author**

1. If the authors have adequately addressed your comments raised in a previous round of review and you feel that this manuscript is now acceptable for publication, you may indicate that here to bypass the “Comments to the Author” section, enter your conflict of interest statement in the “Confidential to Editor” section, and submit your "Accept" recommendation.

Reviewer #1: All comments have been addressed

Reviewer #3: All comments have been addressed

Reviewer #4: (No Response)

2. Is the manuscript technically sound, and do the data support the conclusions?

Reviewer #1: Yes

Reviewer #3: Yes

Reviewer #4: Partly

3. Has the statistical analysis been performed appropriately and rigorously? 

Reviewer #1: Yes

Reviewer #3: Yes

Reviewer #4: N/A

4. Have the authors made all data underlying the findings in their manuscript fully available?

Reviewer #1: Yes

Reviewer #3: Yes

Reviewer #4: Yes

5. Is the manuscript presented in an intelligible fashion and written in standard English?

Reviewer #1: Yes

Reviewer #3: Yes

Reviewer #4: Yes

6. Review Comments to the Author

Reviewer #1: The authors have satisfactorily addressed all the comments and suggestions provided. I appreciate their thorough revisions and attention to detail.

Reviewer #3: The research is sound and also the statistical analysis is done properly.Although the english should be improved like in line 89 " We included adult patients (aged ≥18 years) in our study, targeting individuals identified

as Black, White, Hispanic, or Asian. Measurements" can be replaced by "This study included adult patients (aged ≥18 years), targeting individuals identified

as Black, White, Hispanic, or Asian. Measurements ..." throughout the thesis one should not mention i, we, they etc

Reviewer #4: As most of the questions are asked by other reviewers, I would like to suggest one state of the art conventional approach to the authors:

The descriptive table in page 10 is not rigorous enough. Only median values and IQR are not enough information. I suggest to look into recent literatures for more detailed descriptive statistics. eg. If the outcome is binary then a bivariate table is warranted.

7. PLOS authors have the option to publish the peer review history of their article (what does this mean? ). If published, this will include your full peer review and any attached files.

**Do you want your identity to be public for this peer review?** For information about this choice, including consent withdrawal, please see our Privacy Policy .

Reviewer #1: **Yes: ** Christian Valle Morinaga

Reviewer #3: No

Reviewer #4: No

---

## [Author Response · Author response to Decision Letter 2]

16 Aug 2025

Editorial Office

PLOS ONE

Dear Dr. Emily Chenette and Reviewers,

We are truly grateful for the time and effort you and the reviewers have invested in evaluating our manuscript, "Development and Validation of a Bedside-Available Machine Learning Model to Predict Discrepancies between SaO₂ and SpO₂: Exploring Factors Related to the Discrepancies." We sincerely appreciate your constructive comments and suggestions, which have been invaluable in improving the quality of our work. Please find below our point-by-point responses and the corresponding revisions made to the manuscript.

Reviewer 2

Comment 1:

The manuscript uses first-person pronouns (e.g., “we,” “I”) in several places. The authors should revise the writing style to maintain a formal and objective tone.

Response:

Thank you very much for your valuable comment. In response, we have thoroughly revised the manuscript to eliminate first-person pronouns and ensure a more formal and objective tone throughout the text.

Revision:

No.1: (page 3, para 1)

Original:

“We aimed to explore the factors related to the discrepancy […]”

Revised:

“The aim of this study was to explore the factors related to the discrepancy […]”

No.2: (page 3, para 2)

Original:

“we used non-invasive, readily available bedside information (patient demographics, vital signs, vasopressor use, ventilator use) for developing prediction models with three machine learning methods (decision tree, logistic regression, XGBoost). To make the model accessible, we implemented the model as a web-based application. We further explored the contribution of each variable by using partial dependence plots and SHAP value.”

Revised:

“non-invasive, readily available bedside information (patient demographics, vital signs, vasopressor use, ventilator use) was used to develop prediction models with three machine learning methods (decision tree, logistic regression, XGBoost). To make the model accessible, the model was deployed as a web-based application. Additionally, the contribution of each variable was explored using partial dependence plots and SHAP value.”

No.3: (page 3, para 3)

Original:

“we used 19,804 paired SpO₂ and SaO₂ measurements.”

Revised:

“a total of 19,804 paired SpO₂ and SaO₂ measurements were used.”

No.4: (page 4, para 2)

Original:

“we developed a machine learning model to predict SpO₂–SaO₂ discrepancy and identified vital signs as key contributors. Our findings underscore […]”

Revised:

“a machine learning model was developed to predict SpO₂–SaO₂ discrepancy and identified vital signs as key contributors. These findings underscore […]”

No.5: (page 5, para 2)

Original:

“We aimed to develop and validate […] We further investigated the contribution of each variable to the discrepancy.”

Revised:

“This study aimed to develop and validate […] The contribution of each variable to the discrepancy was further investigated.”

No.6: (page 6, para 1)

Original:

“We employed the eICU database for model development, […]”

Revised:

“The eICU database was used for model development, […]”

No.7: (page 6, para 2)

Original:

“We included adult patients (aged ≥18 years) in our study, […] The difference between SpO₂ and SaO₂ in our study was required to be no more than 20%. We established the following criteria for inclusion: […] Any records with missing data were excluded from our analysis.”

Revised:

“This study included adult patients (aged ≥18 years), […] The difference between SpO₂ and SaO₂ in this study was required to be no more than 20%. The following criteria were established for inclusion: […] Any records with missing data were excluded from this analysis.”

No.8: (page 8, para 1)

Original:

“In the training set (80% random sample), we reduced the risk of algorithmic bias by applying and […] We then selected the model with the highest overall diagnostic value for further fine-tuning and validation, […] we further conducted an exploratory hyper-parameter optimization […] In our study, we employed two distinct techniques for model interpretation: […] Our analyses, visualizations, […]”

Revised:

“In the training set (80% random sample), the risk of algorithmic bias was reduced by applying […] The model with the highest overall diagnostic value was selected for further fine-tuning and validation […] this study conducted an exploratory hyper-parameter optimization […] In this study, two distinct techniques for model interpretation were employed: […] These analyses, visualizations, […]”

No.9: (page 9, para 1)

Original:

“We excluded data points for the following reasons: […] Finally, we identified 4,781 admissions and 19,804 data points suitable for analysis.”

Revised:

“Data points were excluded for the following reasons: […] Finally, 4,781 admissions and 19,804 data points were identified as suitable for analysis.”

No.10: (page 11, para 2)

Original:

“(A) From the eICU dataset (2014–2015), we identified 70,304 data points that included both SpO₂ and SaO₂ measurements. […] (B) From the MIMIC-IV dataset (2008–2019), we identified 81,797 data points that included both SpO₂ and SaO₂ measurements.”

Revised:

“(A) In the eICU dataset (2014–2015), 70,304 data points included both SpO₂ and SaO₂ measurements. […] (B) In the MIMIC-IV dataset (2008–2019), 81,797 data points included both SpO₂ and SaO₂ measurements.”

No.11: (page 12, para 2)

Original:

“After comparing other evaluation scores (S1 Table), we selected the XGBoost model for prediction. Additionally, we constructed a calibration plot to further assess the model’s performance.”

Revised:

“After comparing other evaluation scores (S1 Table), the XGBoost model was selected for prediction. Additionally, a calibration plot was constructed to further assess the model’s performance.”

No.12: (page 13, para 2)

Original:

“We first validated the original XGBoost mode on the MIMIC-IV dataset, […]”

Revised:

“The original XGBoost model was first validated on the MIMIC-IV dataset, […]”

No.13: (page 15, para 2)

Original:

“In this study, we developed a machine learning model using 19,804 data points […]”

Revised:

“In this study, a machine learning model was developed using 19,804 data points […]”

No.14: (page 15 para 3)

Original:

“our model shows lower discrimination.”

Revised:

“this model shows lower discrimination.”

No.15: (page 16, para 2)

Original:

“Our findings about the correlation between the worsening vital signs […]”

Revised:

“These findings about the correlation between the worsening vital signs […]”

No.16: (page 16, para 3)

Original:

“A key strength of our study lies in offering clear insights […]”

Revised:

“A key strength of this study lies in offering clear insight […]”

No.17: (page 17, para 2)

Original:

“Our model demonstrated stable predictive performance […]”

Revised:

“This model demonstrated stable predictive performance […]”

No.18: (page 17, para 3)

Original:

“We have released a lightweight, publicly accessible web application (https://spo2-to-sao2.streamlit.app/) that implements our prediction model.”

Revised:

“A lightweight, publicly accessible web application (https://spo2-to-sao2.streamlit.app/) implementing this prediction model has been released.”

No.19: (page 18, para 1)

Original:

“Our study has several limitations. […] we restricted our data to vital signs and ABG tests that were recorded within 10 minutes. Second, the database we used for training in this study is solely from the United States, which may limit the generalizability of our results. However, to address this concern and enhance the robustness of our findings, we conducted external validation. Moreover, another limitation of our study is the potential bias arising from using multiple data points from the same patients. This aspect is critical in interpreting our data […]”

Revised:

“This study has several limitations. […] the data were restricted to vital signs and ABG tests that were recorded within 10 minutes. Second, the database used for training in this study is solely from the United States, which may limit the generalizability of these results. However, to address this concern and enhance the robustness of these findings, external validation was conducted. Moreover, another limitation of this study is the potential bias arising from using multiple data points from the same patients. This aspect is critical in interpreting the data […]”

No.20: (page 18, para 2)

Original:

“a machine learning model was developed to predict when SpO₂ exceeds SaO₂ by 3% or more, while the prediction ability was suboptimal in a different dataset. We identified vital signs (e.g., temperature and heart rate) as factors associated with these discrepancies. Our findings underscore the awareness for hidden hypoxemia and provide the basis of further study to identify hidden-hypoxia in critically ill patients.”

Revised:

“a machine learning model was developed to predict when SpO₂ exceeds SaO₂ by 3% or more, while the prediction ability was suboptimal in a different dataset. Vital signs (e.g., temperature and heart rate) were identified as factors associated with these discrepancies. These findings underscore the need for awareness of hidden hypoxemia and provide a basis of further studies to identify hidden-hypoxia in critically ill patients.”

Reviewer 4

Comment:

The descriptive table in page 10 is not rigorous enough. Only median values and IQR are not enough information. I suggest to look into recent literatures for more detailed descriptive statistics. If the outcome is binary then a bivariate table is warranted.

Response:

Thank you very much for your constructive comment. In response, we have revised Table 1 to include standard deviations (SD) for continuous variables, as appropriate. Additionally, we have created a new Table 2, which presents a bivariate analysis of each predictor variable stratified by the binary outcome (SpO₂–SaO₂ discrepancy ≥3%). We believe these revisions improve the rigor and clarity of the descriptive statistics in line with recent literature.

We believe these revisions have enhanced the overall quality and clarity of our manuscript and that they adequately address the concerns raised by the reviewers. We are grateful for the valuable feedback and the opportunity to further improve our work. Please do not hesitate to let us know if any additional information or clarification is required.

Best regards,

Raito Sato

School of Medical Sciences, University of Fukui

---

## [Decision Letter · Decision Letter 2]

26 Sep 2025

Development and Validation of a Bedside-Available Machine Learning Model to Predict Discrepancies between SaO₂ and SpO₂: Exploring Factors Related to the Discrepancies

PONE-D-24-58502R2

Dear Dr. Goto,

We’re pleased to inform you that your manuscript has been judged scientifically suitable for publication and will be formally accepted for publication once it meets all outstanding technical requirements.

Kind regards,

Mohsen Mehrabi, Ph.D.

Academic Editor

PLOS ONE

Reviewers' comments:

Reviewer's Responses to Questions

**Comments to the Author**

1. If the authors have adequately addressed your comments raised in a previous round of review and you feel that this manuscript is now acceptable for publication, you may indicate that here to bypass the “Comments to the Author” section, enter your conflict of interest statement in the “Confidential to Editor” section, and submit your "Accept" recommendation.

Reviewer #4: All comments have been addressed

2. Is the manuscript technically sound, and do the data support the conclusions?

Reviewer #4: Yes

3. Has the statistical analysis been performed appropriately and rigorously? 

Reviewer #4: N/A

4. Have the authors made all data underlying the findings in their manuscript fully available?

Reviewer #4: Yes

5. Is the manuscript presented in an intelligible fashion and written in standard English?

Reviewer #4: Yes

6. Review Comments to the Author

Reviewer #4: (No Response)

7. PLOS authors have the option to publish the peer review history of their article (what does this mean? ). If published, this will include your full peer review and any attached files.

**Do you want your identity to be public for this peer review?** For information about this choice, including consent withdrawal, please see our Privacy Policy .

Reviewer #4: No

---

## [Editor Report · Acceptance letter]

PONE-D-24-58502R2

PLOS ONE

Dear Dr. Goto,

I'm pleased to inform you that your manuscript has been deemed suitable for publication in PLOS ONE. Congratulations! Your manuscript is now being handed over to our production team.

Kind regards,

on behalf of

Dr. Mohsen Mehrabi

Academic Editor

PLOS ONE